# One-Pot Method of Synthesizing TEMPO-Oxidized Bacterial Cellulose Nanofibers Using Immobilized TEMPO for Skincare Applications

**DOI:** 10.3390/polym11061044

**Published:** 2019-06-14

**Authors:** Seung-Hyun Jun, Sun-Gyoo Park, Nae-Gyu Kang

**Affiliations:** LG Household and Health Care R&D Center, Seoul 100-859, Korea; junsh@lghnh.com (S.-H.J.); skparke@lghnh.com (S.-G.P.)

**Keywords:** bacterial cellulose, surface modification, TEMPO oxidation, one-pot synthesis, immobilized TEMPO, physical property, skincare

## Abstract

In the skincare field, water-dispersed bacterial cellulose nanofibers synthesized via an oxidation reaction using 2,2,6,6–tetramethyl–1–piperidine–N–oxy radical (TEMPO) as a catalyst are promising bio-based polymers for engineered green materials because of their unique properties when applied to the surface of the skin, such as a high tensile strength, high water-holding capacity, and ability to block harmful substances. However, the conventional method of synthesizing TEMPO-oxidized bacterial cellulose nanofibers (TOCNs) is difficult to scale due to limitations in the centrifuge equipment when treating large amounts of reactant. To address this, we propose a one-pot TOCN synthesis method involving TEMPO immobilized on silica beads that employs simple filtration instead of centrifugation after the oxidation reaction. A comparison of the structural and physical properties of the TOCNs obtained via the proposed and conventional methods found similar properties in each. Therefore, it is anticipated that due to its simplicity, efficiency, and ease of use, the proposed one-pot synthesis method will be employed in production scenarios to prepare production quantities of bio-based polymer nanofibers in various potential industrial applications in the fields of skincare and biomedical research.

## 1. Introduction

Human skin plays an essential role in preventing water loss in the body as it provides the outermost layer of protection from the external environment. In skincare applications, bio-polymers, synthetic polymers, and organic polymers are used to control formulation viscosities, transfer moisture to the skin, increase the stability of the formulation and active ingredient, and protect the skin by forming a coating on its surface [1]. In particular, bio-polymers, such as cellulose, chitosan, and polysaccharide, are known to be skin-friendly substances as they are biocompatible and biodegradable. Among these, cellulose is the most abundant bio-polymer in plants and microorganisms and possesses a number of unique properties that, depending on its origin and the extraction process, allow it to be used in various applications. 

Recently, cellulose nanofibers (CNFs) have attracted wide interest due to their nanoscopic size, ease of preparation, low cost, tunable surface properties, and enhanced mechanical properties, which makes them well suited for use as drug carriers, tissue regenerating scaffolds, water purifying membranes, electrodes, supercapacitors, fluorescent probes, and flexible electronics [2]. In the field of skincare, CNFs have attracted attention as a new potential bio-material with thixotropic properties that allow it to be used for emulsion stabilization, water retention, and rheology modification applications [3].

Bacterial cellulose (BC), which is referred to as bio-cellulose in the skincare field, is a bio-based polymer that is synthesized directly from microorganisms, such as Acetobacter xylinum (*A. xylinum*). BC nanofibers (BCNFs) have a number of advantages over plant-derived cellulose, including a high physical strength, water absorption and retention properties, and a uniform fiber network structure [4,5,6,7]. They are also available in a variety of structural forms, including spheres, gels, sheets, membranes, mats, etc., all of which can be produced by introducing simple modifications into the production strategy. However, a limitation is that it is challenging to use conventional CNF preparation methods to create BCNFs that can be dispersed in water.

CNFs obtained from plant sources can be prepared using chemical, physical, or oxidation methods [8]. Chemical methods typically involve preparing cellulosic nanofibers via the acid digestion of an amorphous area of fibers that, when destroyed, yield nanocrystalline nanofibers. Some physical methods produce CNFs by the mechanical nanofibrillation of chemically purified cellulose pulps using a grinder, high-pressure homogenizer, blender, and high intensity ultrasonicator [8]. Alternatively, aqueous cellulose dispersions of nano-sized structures have been prepared by aqueous counter collision (ACC) using paired water jets without any chemical modification and were then used to convert naturally occurring cellulose fibers into nanofibers [9]. 

In contrast, a higher yield of CNFs can be obtained through the oxidation of cellulose mediated via 2,2,6,6–tetramethyl–1–piperidine–N–oxy radical (TEMPO), which is an oxidation catalyst capable of replacing alcoholic groups of cellulose with aldehyde, ketone, and carboxy groups under mild conditions at room temperature and normal pressure. TEMPO-oxidized cellulose can be dispersed in the aqueous phase by the repulsion force caused by the anionic charge on the surface of the carboxyl groups in the modified cellulose [10,11]. As the resulting material has the advantage of maintaining its physical fiber structure when dispersed in water, its use has been studied in a variety of fields, such as papermaking, membrane filters, heavy metal removal, and cell transfer.

A number of researchers have studied TEMPO immobilized on solid supports, such as polystyrene-bound TEMPO (PS-TEMPO) [12], FibreCat TEMPO [13], TurboBeads-TEMPO (TEMPO immobilized on the outer surface of Fe_3_O_4_ magnetic nanoparticles) [14], and organosilica xerogel (SiliaCat TEMPO) [15,16]. While these types of heterogeneous TEMPO catalyst clouds have been applied to the conversion of molecular chemicals, their use in the form of structured polymers, such as cellulose nanofibers, has been limited due to the steric hindrance between the solid supports and the nanofibers. Recently, Patanker et al. studied nanofibrillated cellulose that had been synthesized via the magnetically separable TEMPO-mediated oxidation and mechanical disintegration of wood pulp [17]. However, to the best of our knowledge, no researchers have yet studied the TEMPO oxidation of bacterial cellulose using immobilized TEMPO as a catalyst.

Previously, we reported the synthesis of TEMPO-oxidized bacterial cellulose nanofibers suitable for use as bio-based polymers for engineered green materials without sodium bromide, which were confirmed to exhibit unique properties on the skin surface [7]. In this case, the TEMPO-oxidized cellulose nanofibers were obtained conventionally using a centrifugation method for washing or drying. However, while these results are promising, it is difficult to implement centrifugation at an industrial scale. In addition, the chemicals used in the drying process, which contain N–oxyl compounds, bromide or iodide, and an oxidizing agent, all remain after drying; thus, this drying method is unsuitable for use in skincare applications.

In contrast, the one-pot synthesis method proposed in this study for producing TOCNs, using TEMPO immobilized on silica beads, is simple, efficient, and easy to use, and the synthesis process can use a simple filtration instead of washing after the cellulose nanofiber dispersion has been prepared. When the TOCNs produced via the one-pot synthesis were applied to the skin surface, they were found to exhibit unique properties similar to those of conventional TOCNs. Thus, they are not only a viable alternative for industrial production for use in the real field, they are also potentially suitable for use in skincare and biomedical research applications.

## 2. Materials and Methods

### 2.1. Materials

A key material in the proposed one-pot process is SiliaCat TEMPO, which is a heterogeneous catalyst/reagent fabricated from a proprietary class of organosilica-entrapped radicals, and which is suitable for the selective oxidation of delicate substrates into higher valued carbonyl derivatives [15]. The SiliaCat TEMPO material used in this study was purchased from SiliCycle Inc. (Quebec, QC, Canada). With regard to the other materials used in the study, sheets of BC were purchased from EZ Costec Co. Ltd. (Gyeonggi, Korea); TEMPO, sodium bromide, and sodium hypochlorite were purchased from Sigma-Aldrich (St Louis, MO, USA), and ethanol was purchased from Daejung Chemical & Metal Co. Ltd. (Gyeonggi, Korea). All other reagents were purchased from Sigma-Aldrich in the highest available commercial grade.

### 2.2. TEMPO Oxidation of BC

The synthesis of TOCNs via the conventional process (C-TOCNs): 20 g of a bacterial cellulose sheet that had been cut into small pieces was suspended in 500 mL of distilled (DI) water containing dissolved sodium hypochlorite and 20 mg of TEMPO catalyst. The oxidation reaction was maintained at a pH of 10 with 0.5 M NaOH. The mixture was vigorously agitated overnight using a magnetic stirrer at 25 °C. The oxidation reaction was then quenched by adding ethanol to the suspension. The products were washed with DI by centrifugation at 10,000× *g* several times, until all of the reactants were completely removed. The final product was stored at room temperature for later use.

The synthesis of TOCNs via the proposed one-pot process (O-TOCNs): the same amounts of bacterial cellulose and sodium hypochlorite were used for the chemical treatment of the process-cellulose. Before SiliaCat-TEMPO was used in the oxidation reaction, it was pretreated to activate the catalyst (Figure 1) by stirring a mixture containing SiliaCat-TEMPO (1 eq.), 4 M HCl (6 eq.) in dioxane, and a 0.5 M solution of N-chlorosuccinimide (5 eq.) in dichloromethane (DCM) was stirred for 15 min. The activated SiliaCat TEMPO was then washed with ethanol and dried completely under vacuum to ensure that all of the reactant had evaporated [15]. An oxidation reaction consisting of bacterial cellulose, sodium hypochlorite, and the activated SiliaCat-TEMPO was facilitated by stirring while the pH was maintained at 10. 

A pure bacterial cellulose aqueous solution was obtained by preparing the aqueous dispersion of cellulose as follows. Ascorbic acid was added to the reacted cellulose solution to neutralize the hypochlorite acid. Then, the SiliaCat-TEMPO was removed by filtering the solution through a vacuum decompression device using a nylon mesh through which only the cellulose nanofiber solution could pass. As a result, the SiliaCat-TEMPO beads remained on the nylon mesh, while the cellulose nanofiber solution passed through to produce a pure nanofiber solution.

### 2.3. Detection of the Sodium Hypochlorite Content 

To neutralize the hypochlorous acid remaining in the reacted cellulose solution, ascorbic acid with a different concentration was added, and the mixture was stirred. Then, a quantitative analysis was conducted using a phosphate buffer solution and N,N–diethyl–p–phenylenediamine (DPD) reagent to determine the amount of free chlorine content. In this test, if residual hypochlorous acid is present, then the color of the reactant will turn red, and its value can be obtained at a wavelength of 510 nm on a spectrophotometer. 

### 2.4. Characterization by Scanning Electron Microscopy 

Oil in water (o/w) emulsions containing TOCNs were prepared, consisting of TOCNs, 1% dimethicone, 3% pentaerythrityl tetraethylhexanoate, 1% hydrogenated polydecene, 7% dipropylene glycol, 7% glycerin, 0.4% methyl glucose sesquistearate, 3% cyclopentasiloxane, 4% xanthan gum, 12% carbomer, 1.5% sorbitol, 0.4% chelating agent, 2% trisodium ethylenediaminetetraacetic acid (EDTA), 2% pH adjusting agent, and water up to 100%. When investigating the surface characteristics, 0.05% of O-TOCNs and C-TOCNs were prepared by casting the solution on porcine skin (1 × 1 cm) and drying at room temperature overnight. These samples were then observed using a field-emission-type scanning electron microscope (FE-SEM; Hitachi S-4000, Tokyo, Japan) after platinum sputtering at 20 mA for 120 s.

### 2.5. Fourier Transform Infrared Spectroscopy (FT-IR) Analysis

The chemical structures of three types of BC samples (pure BC, C-TOCN, and O-TOCN) were analyzed via Fourier transform infrared spectroscopy (PerkinElmer, FT-IR microscope spotlight 200i, Courtaboeuf, France) over a frequency range of 4000 to 450 cm^−1^ and a resolution of 8 cm^−1^.

### 2.6. X-ray Diffractometer (XRD) Analysis 

The XRD patterns of the BC samples were obtained with an X-ray diffractometer (Rigaku, MiniFlex 300/600, Tokyo, Japan) using copper Cu Kα radiation (λ = 1.5406 Å) at 40 kV and 15 mA. The samples were examined with a scanning angle of 2θ from 10° to 80° at a rate of 5°/min, the crystallinity index (CrI) was calculated as a function of the maximum intensity of the diffraction peak from the crystalline region (I_200_) at a 2θ of about 22.5°, and the minimum intensity from the amorphous region (I_am_) at a 2θ angle of about 18° [18]. 

### 2.7. Contact Angle Measurements

The surface wettability of the TOCNs and o/w emulsions containing various concentrations of TOCNs were evaluated by contact angle measurements using a Contact Angle System (OCA, Dataphysics, Filderstadt, Germany) and a high-speed camera. During the measurements, water droplets were deposited directly on the surface of the dried cellulose solution on the porcine skin, and the water contact angles were measured. Three measurements were performed per sample, and the results were averaged. The volume of each water droplet was 10 μL, and each drop was placed using a precision stainless steel tip (Gauge 32, EFD).

## 3. Results and Discussion

### 3.1. One-Pot Synthesis of TOCNs

When synthesizing TOCNs, it is necessary to substitute a certain number of carboxyl groups for those dispersed in the aqueous solution, and the reactants used in the oxidation reaction in the final synthesized TOCNs should be removed. The complete removal of the reactants is especially important in the skincare field. In our previous work, we found that TOCNs synthesized without sodium bromide as a co-oxidant during the conventional TEMPO oxidation reaction did not exhibit significant differences in their physical properties (Appendix A) [7]. 

As a neutralizing agent for sodium hypochlorite, ascorbic acid can completely remove hypochlorite anions and return the solution to its original state without by-products; thus, it is also appropriate for use in skincare applications. To confirm the effect of ascorbic acid for neutralizing hypochlorite anions, the amount of free residual chlorine, dependent on the concentration of ascorbic acid, was measured via the free chlorine DPD method. Note that the available chlorine is present in the form of aqueous molecular chlorine, hypochlorous acid, and hypochlorite ions. The results of the measurements confirmed that the free residual chlorine via the DPD method was completely removed when more than 0.3% of ascorbic acid was added (Figure 2), and it was also confirmed that the sodium hypochlorite could be removed without washing.

Although it was possible to remove the oxidant chemicals, it was difficult to remove the TEMPO catalysts, which is a large problem facing the one-pot synthesis process. During the oxidation reaction of BC using conventional water-soluble TEMPO, repetitive centrifugal separation was found to be the most effective process for removing the TEMPO catalyst. However, when scaling up production for industrial applications, centrifuge equipment does not typically support continuous processing at high levels of relative centrifugal force (RCF) while handing a large amount of reactant. While vacuum filtration can be used to continuously separate TOCN products from a reactant, if the size of the filter pores is too large, they cannot be used to separate the nanofibers. On the other hand, when the size of the filter pores is too small, the reactant does not pass through the filter when the nanofibers are stacked. The large size of the organosilica beads (63–250 μm), which is much larger than the diameter of the TOCNs (30–60 nm), can be selectively separated during the filtration process. Here, the filtration process was conducted using a vacuum decompression device and a nylon mesh (pore size: 50 μm), through which only the cellulose nanofiber solution could pass. In other words, the SiliaCat-TEMPO beads remained on the nylon mesh while the cellulose nanofiber solution passed through, leaving a pure nanofiber solution. Images of the TOCNs taken after filtration showed that the final TOCN solution did not contain silica beads (Appendix A). 

The reaction rate and production yield of the O-TOCNs after the initial oxidation reaction were 72 h and about 80%, while those of the C-TOCNs were 18 h and about 99%, respectively. However, it should be noted that there are several problems with the purification of TOCNs via centrifugation in the conventional process used for industrial applications. For example, low density TOCNs require a high speed centrifugator operating at 13,000 rpm or higher, and the production yield is reduced to less than 70% when the process is repeated more than five times, as is the case when a tubular type centrifuge is used for treating the bulk scale product. Second, the treatment of large amounts of TOBCNs via centrifugation is challenging due to the lack of automated equipment that is capable of continuous centrifugation. In contrast, the proposed one-pot synthesis method is a viable alternative that does not suffer from these limitations. Moreover, other reactants such as sodium bromide and sodium hypochlorite were used for the conventional synthesis of TOCNs. The complete removal of the reactants is especially important in the skincare field. Our proposed process is a very efficient way to remove all reactants without any additional process.

### 3.2. Characterization of the TOCNs Obtained via One-Pot Synthesis

It was confirmed that the properties of the TOCNs obtained after the one-pot synthesis (O-TOCNs) were different from those of the TOCNs obtained via the conventional synthesis method (C-TOCNs). The carboxyl and aldehyde contents were measured using the conductivity titration method and found to be 1.02 ± 0.08 mmol g^−1^ and 0.12 ± 0.03 mmol g^−1^, respectively, for the O-TOCNs, and 1.12 ± 0.10 mmol g^−1^ and 0.08 ± 0.04 mmol g^−1^, respectively, for the C-TOCNs. The carboxyl and aldehyde contents of pure BC were 0.02 ± 0.01 mmol g^−1^ and 0.00 mmol g^−1^, respectively. The carboxyl and aldehyde groups of both of the TOCNs were increased compared to those of the original BC, but those of the two samples were found to follow a similar trend. From the SEM images in Figure 3, it can be seen that the average diameter of the BC nanofibers was approximately 100 nm, decreasing to 30–50 nm after oxidation in the conventional process. However, the average diameter of the TOCNs was about 80 nm, which is closer to the diameter of the original BC nanofibers than that of the C-TOCNs (Figure 3). This implies that when soluble TEMPO is used, the ionic charge between the fibrils that bundles them into fibers is weakened. In contrast, the immobilized TEMPO catalyst does not penetrate the fiber bundle but only acts on the surface of the BC fiber. To confirm whether the O-TOCNs were structurally different from other water-soluble biopolymers, the nanometer-resolution structures of the bio-polymers were observed via high-resolution imaging provided by an FE-SEM (Appendix A). Carbopol (polyacrylic acid), xanthan gum (polysaccharide), and carboxymethyl cellulose (CMC) were selected for comparison purposes as water-soluble bio-polymers that are commonly used in the skincare field. When compared, the results showed that while the other bio-polymers did not exhibit any particular structure, the TOCNs had fiber structures in which bundles of nanofibers were entangled. This suggests that, as water-dispersed biopolymers with a unique fiber structure, the synthesized TOCNs will likely maintain their characteristics on the surface of the skin.

The FT-IR spectra of the three types of BC samples are shown in Figure 4. As can be seen in the figure, the pure BC had no obvious absorption band from 1500 to 2000 cm^−1^ (Figure 4a); in Figure 4b,c, the signal at 1603 cm^-1^ attributed to the carboxyl groups was consistent with that of the carboxyl-functionalized BC, which indicates that the hydroxyl groups at the C6 position in the BC had been successfully converted into carboxyl groups [19]. 

The XRD patterns of the pure BC and two types of TOCNs exhibited 2θ diffraction peaks at 14.5°, 16.4°, and 22.5°, usually attributed to the 101 (amorphous region), 10 (amorphous region), and 200 (crystalline region) crystallographic planes, respectively (Figure 5) [20]. These represent the primary diffraction of the (1–10), (110), and (200) crystal planes, respectively, which elucidates the structure of the cellulose I crystal. After oxidation via different methods, the diffraction peaks showed a slightly higher intensity compared to those of the BC, which indicates that the TOCNs retained the crystal structure of cellulose I with an unexpected small improvement in the crystallinity [18]. In other words, the oxidation process was shown to have little impact on the crystal structure of the BC. The crystallinity of the pure BC, C-TOCNs, and O-TOCNs were 75.7%, 79.5%, and 75.3%, respectively, which indicates that the crystallinity of the two TOCNs was similar. 

### 3.3. Properties of O-TOCNs on the Surface of Skin

In a previous study, it was found that the nanofibrous structure of TOCNs give them unique physical properties, such as a water-absorbing capacity, elasticity, and the ability to block particulate matter (PM) on the skin surface [7]. Here, it was confirmed that the one-pot synthesized TOCNs exhibited similar properties on the skin surface with respect to their water-absorbing capacity and blockage of PM on the skin. A water droplet was placed on porcine skin covered with O-TOCNs in an o/w emulsion containing different concentrations of O-TOCNs, and the changes in the water contact angle over time were measured. As shown in Figure 6a, the contact angle of the water over time was greatly reduced as the TOCNs’ content in the o/w emulsion increased. These results suggest that the high carboxylate content of the TOCNs contributes to their excellent water-absorbing ability, resulting in a low resistance to water along with their other unique properties. It also indicates that the TOCNs have a water-absorption capacity, even when mixed with other materials. In the experiment conducted to evaluate the removal of PM on the skin, the samples (a combination of water and o/w emulsions with and without TOCNs) were applied on a 2 × 2 cm area on the upper arm, along with 8 μL of a carbon black solution (1.25 wt %), and they were then dried. The area was rubbed ten times with a small amount of water before being washed with tap water. The images before and after the application of carbon black were taken with a magnifier and analyzed with the Image Pro Plus software package (Media Cybernetics). We previously reported that TOCNs, which form a network of nanofibers on the skin surface, play a role in blocking particular matter. In this test, the o/w emulsion containing TOCNs was found to prevent the carbon black from entering into the microgrooves on the surface of the skin, and thereby allowed it to be easily removed (Figure 6b).

## 4. Conclusions

A one-pot synthesis of water-dispersed bacterial cellulose nanofibers was prepared using a process that combined the neutralization of sodium hypochlorite via the addition of ascorbic acid and a filtration process using TEMPO that was immobilized on silica beads. The TOCNs obtained after the one-pot synthesis were found to have nanofibrous structures with the unique properties of bacterial cellulose, a water-absorbing capacity, and the ability to remove PM from the surface of skin. Based on our results, our proposed one-pot synthesis method for TOCNs is simple, efficient, and easy to use, and is therefore ideally suited to the preparation of bio-based polymer nanofibers in production quantities. It is therefore anticipated to have potential industrial applications in the skincare and biomedical research fields.

## Figures and Tables

**Figure 1 polymers-11-01044-f001:**
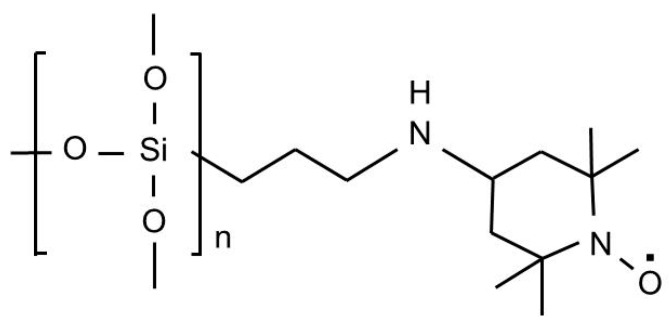
Schematic showing the structure of the immobilized 2,2,6,6–tetramethyl–1–piperidine–N–oxy radical (TEMPO) on silica beads.

**Figure 2 polymers-11-01044-f002:**
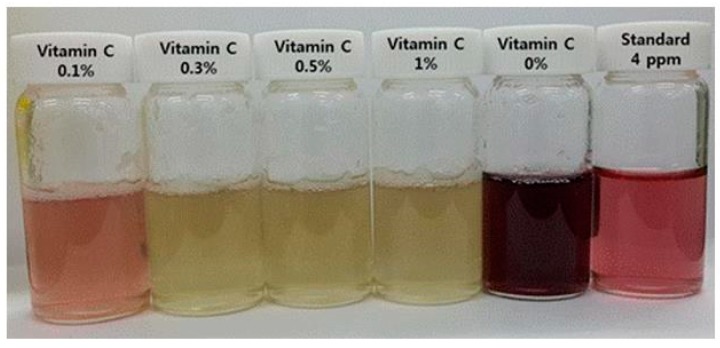
Neutralization of the sodium hypochlorite using ascorbic acid.

**Figure 3 polymers-11-01044-f003:**
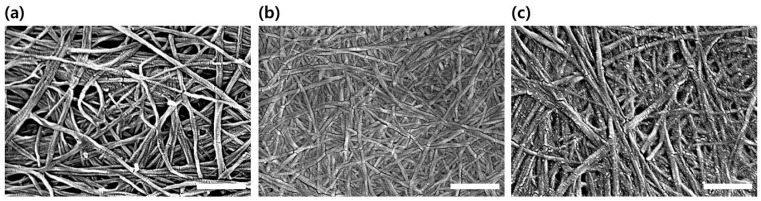
SEM images of (**a**) pure BC, (**b**) C-TOCNs and (**c**) O-TOCNs. Scale bar = 500 nm.

**Figure 4 polymers-11-01044-f004:**
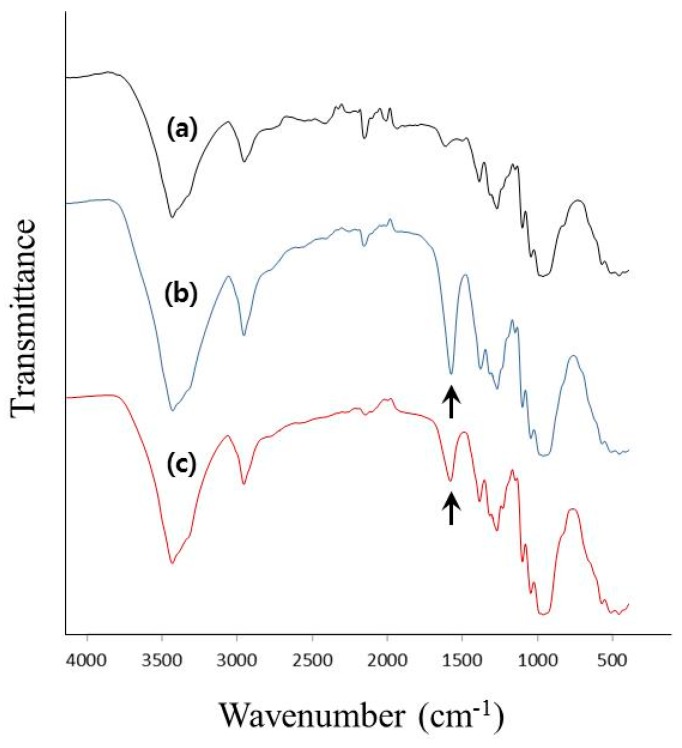
The FT-IR spectra of (**a**) pure BC, (**b**) C-TOCNs and (**c**) O-TOCNs.

**Figure 5 polymers-11-01044-f005:**
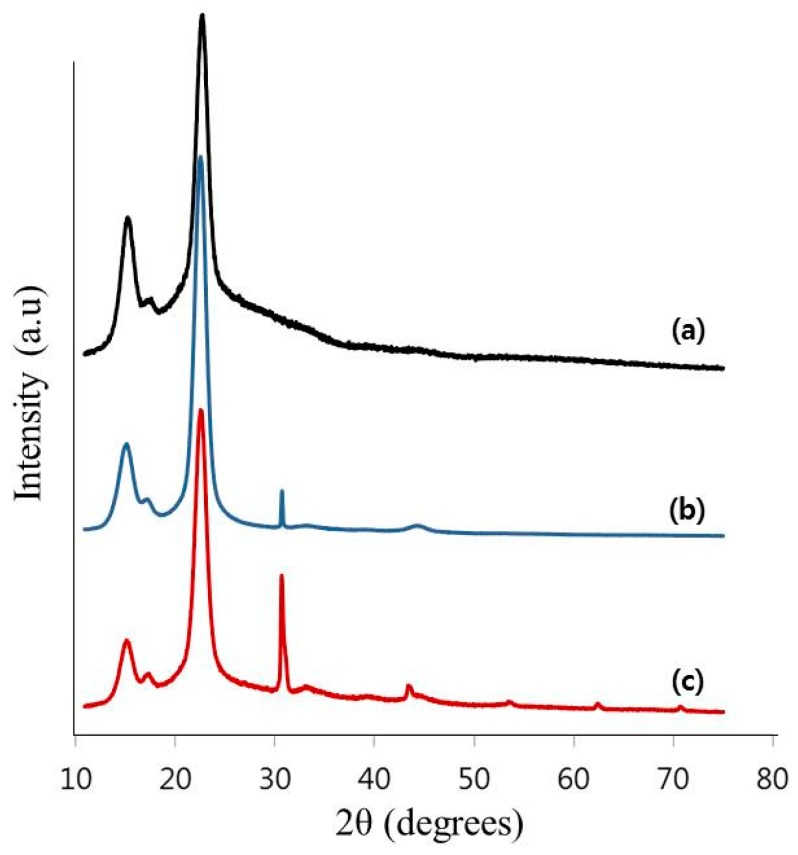
The XRD patterns of (**a**) pure BC, (**b**) C-TOCNs and (**c**) O-TOCNs.

**Figure 6 polymers-11-01044-f006:**
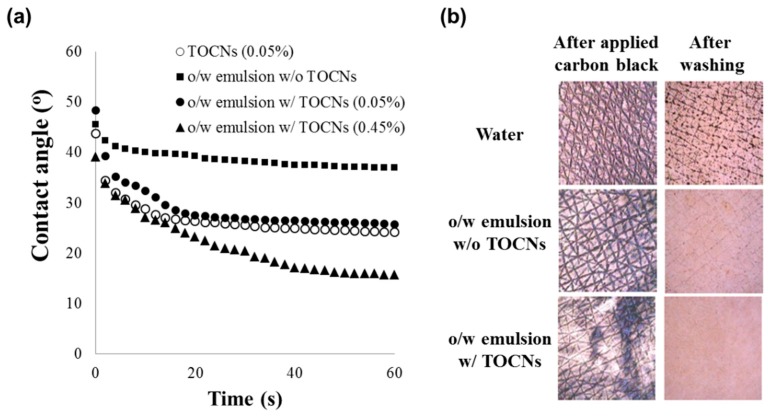
(**a**) Changes in the contact angle of various concentrations of TOCNs in o/w emulsion over time and (**b**) the washing experiment conducted to remove the carbon black from the skin.

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
