# Peer review of "One-Pot Method of Synthesizing TEMPO-Oxidized Bacterial Cellulose Nanofibers Using Immobilized TEMPO for Skincare Applications"

_polymers, 2019, doi:10.3390/polym11061044_

Round 1

Reviewer 1 Report

All the issues that I raised in previous iteration (ms. ID polymers-491968) have been addressed. The new text and characterization data (FTIR,  XRPD and new SEM micrographs in Fig. 3) added to the ms. (I assume that in response to another reviewer's comments) have improved the quality of the final ms. No further changes are requested.

Author Response

All the issues that I raised in previous iteration (ms. ID polymers-491968) have been addressed. The new text and characterization data (FTIR,  XRPD and new SEM micrographs in Fig. 3) added to the ms. (I assume that in response to another reviewer's comments) have improved the quality of the final ms. No further changes are requested.

[Response]

We deeply appreciate the reviewer’s invaluable comments to improve the quality of our manuscript in previous revision.

Reviewer 2 Report

Although the authors have made a lot of revisions and supplements to the paper according to the reviewers' opinions, there are still many aspects that need to be further improved:

 1. The innovation of the paper is not well demonstrated.

 2. The characterization of nanocellulose itself is not detailed.

 3. In Figure 1 of line 143, there is no polystyrene in the catalyst schematic structure. Why do polystyrene beads still appear in lines 138 and 229?

 4. Why are the two figures in the supplementary file deleted from the original text, and what is the significance of doing so?

Author Response

Although the authors have made a lot of revisions and supplements to the paper according to the reviewers' opinions, there are still many aspects that need to be further improved:

 1. The innovation of the paper is not well demonstrated.

[Response]

For a fiber structure, such as that of TOCNs, to exhibit the unique functions on the skin surface, the inherent function of the cellulose before synthesis should be excellent, uniform structure of fibers should be synthesized, and the network consisting of cellulose nanofibers should be able to cover the skin uniformly. In the case of plant cellulose, a top-down process, such as homogenization or fibrillation, is required to produce nanometer size, which is likely to randomly form fiber lengths or diameters.

On the contrary, BC has high purity in contrast to plant cellulose because of its production from bacteria directly. Moreover, BC exhibits unique physical, chemical, and mechanical properties including high crystallinity, large surface area, elasticity, and biocompatibility. This is reason to select the TOCNs from BC as excellent biomaterials for the skin tissue in our previous study. However, it is much more difficult to conduct the washing process via repetitive centrifugation than by the pretreatment of the immobilized catalyst during the scale-up process. Moreover, other reactants such as sodium bromide and sodium hypochlorite were used for conventional synthesis of TOCNs. The complete removal of the reactants is especially important in the skincare field. Our proposed process is a very efficient way to remove all reactants without any additional process. For this reason, we presented a one-pot synthesis method as a viable alternative to existing processes for use of real-field.

The following description was inserted in the main text of the revised manuscript.

[Page 6, line 242 – page 6, line 245]

Moreover, other reactants such as sodium bromide and sodium hypochlorite were used for conventional synthesis of TOCNs. The complete removal of the reactants is especially important in the skincare field. Our proposed process is a very efficient way to remove all reactants without any additional process.

 2. The characterization of nanocellulose itself is not detailed.

[Response]

We have provided the characterization of nanocellulose (pure bacterial cellulose) such as SEM, FT-IR, XRD analysis. We newly prepared the carboxy and aldehyde contents of pure bacterial cellulose as followings.

[Page 7, line 253 – page 7, line 256]

The carboxyl and aldehyde content of pure BC were 0.02 ± 0.01 mmol g−1 and 0.00 mmol g−1, respectively. The carboxyl and aldehyde groups of both the TOCNs were increased compared to those of the original BC, but those of the two samples were found to follow a similar trend.

 3. In Figure 1 of line 143, there is no polystyrene in the catalyst schematic structure. Why do polystyrene beads still appear in lines 138 and 229?

[Response]

That was a mistake on our part and has now been corrected in the manuscript.

 4. Why are the two figures in the supplementary file deleted from the original text, and what is the significance of doing so?

[Response]

In reviewers' comments on previous manuscript, they commented that these data are not enough valuable information. However, we think that figure s2 and s3 were considered necessary for this paper as following reasons.

In case of Figure S2, it was considered appropriate to place the supporting information in the back up data indicating that the immobilized TEMPO catalyst remained on the nylon mesh while the cellulose nanofiber solution passed through, leaving a pure nanofiber solution.

In case of Figure S3, TOCNs were water-dispersed but retains the fiber structure unlike other polymers. As a result of zoom-in image analysis, the results showed that while the other bio-polymers did not exhibit any particular structure, the TOCNs had fiber structures in which bundles of nanofibers were entangled. This suggests that as water-dispersed biopolymers with a unique fiber structure, the synthesized TOCNs will likely maintain their characteristics on the surface of the skin.

Round 2

Reviewer 2 Report

After several serious revisions and improvements by the author, and detailed replies to the reviewer's questions, the paper can be published after careful proof of charts  and writing.

This manuscript is a resubmission of an earlier submission. The following is a list of the peer review reports and author responses from that submission.

Round 1

Reviewer 1 Report

The following issues need to be addressed:

Title needs to be shortened, avoiding repetition of TEMPO. Please re-write as: "One-pot method of synthesizing oxidized bacterial cellulose nanofibers using immobilized TEMPO for skincare applications".

Line 103: SiliaCat-TEMPO (capital C)

Line 116: Figure 1 caption needs to be updated. "Schematic showing the structure of SiliaCat TEMPO heterogeneous catalysts (R723-100)". Moreover, the Si circle makes no sense. Please replace with the usual schematic (e.g. https://www.silicycle.com/media/catalog/product/cache/1/image/265x150/9df78eab33525d08d6e5fb8d27136e95/s/i/siliacat-tempo_1.png), re-drawn in ChemDraw.

Lines 149 to 151: Please delete Figure 2, which add no value to the ms.

Lines 153-158: Delete paragraph: "Sodium... [13-15]."

Line 168: Delete equations 1 and 2 (and reference in line 160), which are not accurate (strictly, sodium hypochlorite should appear instead of hypochlorous acid, but -as noted by the authors- it is a mixture, and the reaction would be confusing for the reader).
Lines 181-183: The paragraph "SiliaCat... [12]" should appear in line 103, not here.

Reviewer 2 Report

Seung-Hyun Jun and colleagues describe One-pot method of synthesizing TEMPO-oxidized bacterial cellulose nanofibers using immobilized TEMPO for skincare applications (polymers-491968). The contents are not rich, the research methods and technique are not innovative enough, the data demonstration and conclusion are not rigorous. The quality of papers cannot meet the requirements of reviewing manuscripts. 

1. The content shown in the title and abstract of the paper does not correspond to the actual research content of the paper. The title and abstract expressed to the reader is to prepare bacterial nanocellulose in one-pot using immobilized TEMPO as reagent. Therefore, the author should discuss the yield of nanocellulose, the structure of nanocellulose (such as size, aspect ratio, crystallinity, etc.), the physical properties of nanocellulose membranes, and the similarities and differences of residues in nanocellulose products obtained by different methods, rather than simply comparing the application of nanocellulose in skin care products.

2. The preparation of nanocellulose by commercial SiliaCat TEMPO reagent has not been reviewed and cited in this paper. Where is innovation?

3. The characterization of the paper is not rich enough. Fig. 4, Fig. 5 (SEM) and Fig. 6 (FE-SEM) are not clear enough. These graphs do not express enough valuable information.

4. The contents about detection of the sodium hypochloride are no need to discuss it in detail.

5. Line111-113, what is PS-TEMPO? Where did polystyrene beads come from?

6. According to the paper, SiliaCat #TEMPO needs to be activated before use, and the activated process is complex. Organic reagents such as dichloromethane, dioxane and N-chlorosuccinimide are used. Therefore, the method is not simple and environmentally friendly.